# Emerging Insights into the Relationship Between Amino Acid Metabolism and Diabetic Cardiomyopathy

**DOI:** 10.3390/biom15070916

**Published:** 2025-06-22

**Authors:** Yi Wen, Xiaozhu Ma, Shuai Mei, Qidamugai Wuyun, Jiangtao Yan

**Affiliations:** 1Department of Cardiology, Division of Internal Medicine, Tongji Hospital, Tongji Medical College, Huazhong University of Science & Technology, Wuhan 430030, China; wenyi@hust.edu.cn (Y.W.); xzma2023@hust.edu.cn (X.M.); moshine@hust.edu.cn (S.M.); d202282253@hust.edu.cn (Q.W.); 2Hubei Key Laboratory of Genetics and Molecular Mechanisms of Cardiological Disorders, Wuhan 430030, China

**Keywords:** amino acid metabolism, diabetic cardiomyopathy, branched-chain amino acid, aromatic amino acid

## Abstract

Diabetes mellitus (DM) is a complex global pandemic that frequently leads to multiple complications. Diabetic cardiomyopathy (DCM) is the primary cause of heart failure in patients with type 1 and 2 diabetes and is fundamentally characterized by abnormalities in myocardial structure and function. Metabolic disorders occupy a leading role in the pathogenesis of DCM, manifesting as disrupted substrate metabolism, dysregulated signaling pathways, and energy imbalance. Given the limited benefits of conventional therapeutic strategies targeting glucolipid metabolism, increasing research efforts have focused on amino acid metabolism. Amino acids are involved in the synthesis of nitrogen-containing compounds and serve as an energy source under specific conditions. Moreover, emerging studies demonstrate that metabolic disturbances of specific amino acids—such as branched-chain amino acids (BCAAs), glutamine, and arginine—exacerbate mitochondrial dysfunction and oxidative stress, thereby promoting myocardial fibrosis and cardiomyocyte injury. Therefore, this review aims to summarize the general characteristics and regulatory pathways of amino acid metabolism, as well as the specific mechanisms by which metabolic alterations of amino acids contribute to the pathogenesis and progression of diabetic cardiomyopathy, with the hope of advancing more effective translational therapeutic approaches.

## 1. Introduction

Diabetes mellitus (DM), a prevalent metabolic disorder, has demonstrated a marked increase in global adult prevalence from 7% in 1990 to 14% in 2022, as evidenced by data from the Non-Communicable Diseases Risk Factor Collaboration (NCD-RisC) [1], highlighting that diabetes continues to be a significant global public health challenge. Cardiovascular disease (CVD) is the primary cause of morbidity and mortality in individuals with diabetes [2], with their risk being 2 to 4 times greater than that of individuals with normal blood glucose levels [3]. Diabetic cardiomyopathy (DCM), initially characterized by Rubler et al. in 1972, is defined as a condition of heart failure that occurs in the absence of coronary artery disease, hypertension, or valvular heart disease [4]. In June 2024, the European Society of Cardiology (ESC) expert consensus proposed a newly revised definition of DCM, characterized by myocardial systolic and/or diastolic dysfunction in the presence of diabetes, irrespective of other coexisting risk factors [5]. The updated definition significantly broadens the scope of diabetic cardiomyopathy, facilitating early detection and standardized interventions, thereby delaying heart failure progression and reducing morbidity and mortality.

Metabolic disorders, associated with disruptions in biochemical processes governing energy conversion and nutrient utilization, are fundamentally driven by the metabolic dysregulation of glucose, lipid and amino acid and frequently progress to chronic diseases, including diabetes mellitus and cardiovascular complications. Although glycemic control and heart failure medications exert beneficial effects on these conditions, currently no specific therapeutic approaches exist for targeting diabetic cardiomyopathy. Over the past decades, the pathophysiology of DCM has been preliminarily elucidated [6], with emerging evidence identifying the regulation of amino acid metabolism as a vital mediator and therapeutic target for invention in diabetes [7,8,9] and concurrent cardiovascular risk [10,11,12,13]. Therefore, given the current limitations in effective diagnostic and therapeutic methods [6] and these insightful clinical findings, we aim to systematically summarize the general principles and regulatory mechanisms of amino acid metabolism and review the critical role of amino acid metabolism in the development, diagnosis, treatment, and prognosis of diabetic cardiomyopathy.

## 2. Overview of Amino Acids and Their Metabolism

Approximately 20 amino acids participate in human protein synthesis, including essential amino acids (EAAs), which must be obtained from the diet, and non-essential amino acids (NEAAs), which are synthesized endogenously (Table 1). Beyond serving as protein building blocks, amino acids play critical roles in metabolic regulation, signaling, and immune function. Structurally, leucine, valine, and isoleucine are classified as branched-chain amino acids (BCAAs), while tryptophan, phenylalanine, and tyrosine are aromatic amino acids (AAAs). Amino acids are derived from both endogenous protein turnover and dietary intake, forming a metabolic pool that supports both anabolic processes (e.g., NEAA synthesis, protein formation) and catabolic pathways (e.g., transamination, deamination, and nitrogen metabolism).

## 3. Amino Acid Anabolism

Endogenous synthesis of non-essential amino acids (NEAAs) primarily relies on intermediates from glycolysis and the tricarboxylic acid (TCA) cycle, such as 3-phosphoglycerate and oxaloacetate. These substrates undergo transamination and other enzymatic reactions to yield NEAAs, a process essential for amino acid homeostasis and the conservation of essential amino acids (EAAs), particularly under stress conditions such as diabetic cardiomyopathy [14,15,16].

In contrast, dietary intake is the main source of EAAs for the human body. Intestinal absorption is mediated by various solute carrier (SLC) family transporters expressed on the apical membranes of epithelial cells. These amino acid transporters (AATs) include B^0^AT1 (SLC6A19) for neutral amino acids, b^0,+^AT (SLC7A9) for cationic amino acids, EAAT3 (SLC1A1) for anionic amino acids, PAT1 (SLC36A1) for glycine and proline, as well as β-amino acids, PepT1 (SLC15A1) for dipeptides and tripeptides, SIT (SLC6A20) for glycine and proline, and TauT (SLC6A6) for β-amino acids [17] Additionally, the large neutral amino acid transporter (LAT) family, which is also part of the SLC superfamily, is expressed on the basolateral membrane of intestinal epithelial cells and plays a crucial role in the coordination of amino acid excretion and reabsorption across the epithelium [18,19]. The antiporter LAT2 (SLC7A8) facilitates the transport of all neutral amino acids except proline [20], and y^+^LAT1 (SLC7A7) transports both cationic and neutral amino acids [21]. The uniporters TAT1 (SLC16A10) and LAT4 (SLC43A2) are responsible for the transport of AAAs and BCAAs, respectively [17] (LAT4 is also believed to participate in the transport of methionine and phenylalanine [22,23]).

In addition to dietary intake, EAA supplementation is vital in specific physiological or pathological states. For instance, lysine is used to support pediatric growth [24], and EAA mixtures combined with resistance training improve muscle function in sarcopenia [25]. Imbalanced EAA intake may contribute to conditions such as hepatic steatosis [26], neuronal dysfunction [27], or renal damage via excessive tryptophan metabolism [28]. Conversely, BCAA restriction is essential in disorders like maple syrup urine disease (MSUD) [29].

## 4. Amino Acid Catabolism

The degradation routes of amino acids exhibit both similarities and distinct characteristics. Amino acids absorbed through digestion, including alanine and aromatic amino acids, are metabolized predominantly in the liver. In contrast, the catabolism of BCAAs primarily occurs in skeletal muscle and affects their function simultaneously [30] Furthermore, several amino acid-degrading enzymes (AADEs), notably branched-chain aminotransferase (BCAT) and branched-chain keto acid dehydrogenase (BCKDH), are expressed in human intestinal epithelial cells [31], indicating that dietary amino acids, particularly BCAAs, can undergo degradation within the intestinal environment [32] (Table 1 and Figure 1).

### 4.1. Deamination of Amino Acids

Amino acid catabolism primarily occurs through deamination, with most amino acids undergoing trans-deamination catalyzed by aminotransferases and *L*-glutamate dehydrogenase, producing ammonia and α-keto acids—glutamate serving as a central intermediate. In the liver and kidney, some amino acids are further catabolized by *L*-amino acid oxidase, generating α-keto acids, H_2_O_2_, and ammonia. In skeletal and cardiac muscle, where glutamate dehydrogenase is less active, the purine nucleotide cycle (PNC) predominates. Certain amino acids such as histidine [33] and phenylalanine [34] also undergo non-oxidative deamination by gut microbes. The resulting α-keto acids can be funneled into energy production, NEAA synthesis, or converted into glucose and lipids via the TCA cycle. Notably, branched-chain α-keto acids (BCKAs) have been linked to tumor progression [35] and myocardial ischemia-reperfusion injury [36].

### 4.2. Metabolism of Blood Ammonia

The deamination of amino acids is the primary source of blood ammonia. In hepatocytes, glutamine (one of the carriers of ammonia) is transported into the mitochondrial matrix via the SLC1A5 transporter and hydrolyzed by glutaminase to produce glutamate and one molecule of ammonia. Thereafter, glutamate is further deaminated by *L*-glutamate dehydrogenase to yield α-ketoglutarate and another molecule of ammonia [37]. In the cytoplasm, alanine aminotransferase catalyzes the conversion of alanine and α-ketoglutarate to pyruvate and glutamate. Via carriers expressed on the inner mitochondrial membrane, including aspartate/glutamate carrier 2 (AGC2/SLC25A13) and glutamate carriers (GC1/SLC25A22 and GC2/SLC25A18) [38], glutamate is subsequently transported into the mitochondrial matrix. Ammonia can participate in the synthesis of NEAAs and other nitrogen-containing compounds, contributing to the maintenance of systemic ammonia homeostasis.

### 4.3. Metabolism of Individual Amino Acids

In addition to deamination and the metabolism of blood ammonia mentioned above, certain amino acids are involved in specialized metabolic pathways with significant physiological implications.

#### 4.3.1. Aromatic Amino Acids

Under the catalysis of oxygenase, aromatic amino acids (AAAs) undergo decomposition to produce catecholamines, melanin, and multiple intermediates in glucose and lipid metabolism. Aromatic amino acid decarboxylase (AADC) mediates the production of aromatic amines by the gut microbiota and has been confirmed to stimulate colonic 5-hydroxytryptamine (5-HT) synthesis [39]. In addition to synthesizing proteins, tryptophan can be metabolized into kynurenine, serotonin (5-HT), indole derivatives, and nicotinic acid. Metabolites and enzymes involved in the kynurenine pathway (KP) have been implicated in lifespan extension [40], diabetic nephropathy [41], neuropsychiatric disorders [42], immune regulation [43], tolerance [44], and drug addiction [45].

#### 4.3.2. Histidine

Histidine decarboxylation yields histamine; blockade of its receptors H_1_R/H_2_R exacerbates inflammatory immune responses via the NLRP3/caspase-1 proinflammatory cytokine pathway [46].

#### 4.3.3. Serine

Serine hydroxymethyltransferase 2 (SHMT2) catalyzes the conversion of serine to glycine, generating a one-carbon unit that plays critical roles in development, immunity, and tumorigenesis. Additionally, one-carbon units support the synthesis of S-adenosylmethionine (SAM), the primary cellular methyl donor, which is closely associated with energy metabolism and epigenetic regulation [47,48].

#### 4.3.4. Cysteine and Cystine

Cysteine and cystine are two other sulfur-containing amino acids that can interconvert with each other. Through a series of enzymatic reactions, cysteine generates endogenous hydrogen sulfide (H_2_S) and active sulfate (PAPS), which regulate vascular function, oxidative stress, inflammation, tumor proliferation, hormone homeostasis, and renal excretion [49].

#### 4.3.5. Arginine

Arginine, under the catalysis of nitric oxide synthase (NOS), produces nitric oxide (NO), the first gasotransmitter discovered with complex cellular signaling functions. Over four decades of multidisciplinary research have established the pivotal role of NO in cardiovascular function, metabolism, neurotransmission, and immune modulation [50].

#### 4.3.6. Branched-Chain Amino Acids

Following transportation into the mitochondria via the SLC25A44 carrier, BCAAs undergo transamination by branched-chain aminotransferase 2 (BCAT2) and oxidative decarboxylation by BCKDH, which generates BCKAs that participate in the TCA cycle and produce significant intermediates such as 3-hydroxyisobutyrate (3-HIB) and monomethyl branched-chain fatty acids (mmBCFAs) [51].

## 5. Regulation of Amino Acid Metabolism

Amino acid homeostasis refers to the dynamic equilibrium of amino acids and related metabolites within the body, which is the foundation for the proper functioning of protein synthesis, energy metabolism, and other vital biological processes. In addition to the regulation by enzymes (Figure 2), diet, signaling pathways, and amino acid transporters discussed below, amino acid metabolism is influenced to varying degrees by circadian rhythm [52], emotional health [53], and even climate change [54].

### 5.1. Amino Acid Metabolic Enzymes

As cellular biomolecules with catalytic capabilities, enzymes, particularly key enzymes, play a pivotal role in amino acid metabolism. For example, ATP-binding cassette (ABC) transporter proteins are among the rate-limiting factors in the kynurenine pathway (KP) of tryptophan, modulating the availability of tryptophan as a substrate for another rate-limiting enzyme, tryptophan-2,3-dioxygenase (TDO) [55]. Plant-derived bioactive compounds, such as quercetin [56], naringenin and naringin [57], along with organic acids found in hawthorn [58], can influence the activity of digestive enzymes that act on proteins in the gastrointestinal tract. Various vitamins, especially B vitamins, serve as coenzymes or cofactors and regulate the functional activity of enzymes involved in amino acid metabolism. According to the review of Nieraad et al., deficiencies in one or all vitamin B6, vitamin B12, and folic acid, due to factors such as inadequate intake, increased demand, or medication side effects, may lead to metabolic disturbances in one-carbon units and the incidence of hyperhomocysteinemia (HHCy) [59].

The transcription of amino acid-degrading enzymes (AADEs), which are predominantly localized in the liver and exhibit substrate specificity, is regulated by multiple factors, including diet, the gut microbiota, hormonal signaling, and transcription factors [60]. High-protein diets increase the activity and mRNA expression of amino acid-related metabolic enzymes in the liver while concurrently downregulating the expression of enzymes and mRNAs involved in carbohydrate and lipid metabolism. These alterations may be attributed to the utilization of excess amino acids as an energy source, with AMP-activated protein kinase (AMPK) as a potential regulatory factor [61,62,63]. Research has demonstrated that certain hormones, such as glucagon, corticosteroids, growth hormone, and insulin-like growth factor-1 (IGF-1), are involved in both the transcriptional and non-transcriptional regulation of amino acid metabolic enzymes, thereby influencing processes such as amino acid catabolism and the urea cycle [64,65,66] Moreover, several transcription factors contribute to this regulatory network. For example, zinc finger and BTB domain-containing protein 1 (ZBTB1) [67] and Krüppel-like factor (KLF6) [68] regulate the expression of asparagine synthetase (ASNS) and enzymes involved in the metabolism of BCAAs, particularly branched-chain keto acid dehydrogenase E1 subunit beta (BCKDHB). Additionally, Warnhoff et al. elucidated the role of hypoxia-inducible factor-1 (HIF-1) in the negative feedback regulation of cysteine homeostasis. In brief, the elevated level of cysteine promotes the generation of H_2_S signals, which subsequently stimulate HIF-1-mediated transcription of *cdo-1* via the *rhy-1/cysl-1/egl-9* signaling pathway, thereby enhancing the cysteine degradation process catalyzed by cysteine dioxygenase (CDO-1) [69].

### 5.2. Diet

Dietary composition plays a pivotal role in modulating amino acid metabolism. Both the source [70] and quantity [71] of dietary protein influence amino acid digestibility and absorption. Plant-derived proteins typically exhibit lower digestibility due to intrinsic factors such as high proline content, low solubility, and dense peptide structures, as well as extrinsic factors like antinutritional compounds and physical barriers [72] The absorption capacity of the small intestine is limited, and excess dietary proteins that reach the large intestine undergo fermentation into nitrogenous compounds and other metabolites [73]. Furthermore, diets rich in amylose significantly upregulate mRNA levels of amino acid transporters while downregulating amino acid-degrading enzymes (AADEs) in the ileum. These changes are accompanied by increased activation of the mTOR signaling pathway, as indicated by elevated levels of p-mTOR, p-4EBP1, and p-S6K1, thereby reducing amino acid consumption within the gut and promoting their systemic availability for protein synthesis [74] Compared with high-fat or high-protein diets, high-carbohydrate diets increase metabolic efficiency in amino acid utilization, an effect associated with changes in butyrate production [75]. These findings suggest that dietary structure can modulate amino acid metabolism and systemic homeostasis, with potential implications for metabolic disorders such as diabetic cardiomyopathy.

### 5.3. Amino Acid Transporters

Amino acid transporters (AATs) are regulated by a diverse array of transcription factors. Transcription factors that modulate the expression of excitatory amino acid transporter 1 (EAAT1) can be categorized into positive regulators (such as NF-κB, CREB, and β-catenin) and negative regulators (such as N-myc and YY1) [76] For excitatory amino acid transporter 2 (EAAT2), NF-κB and REST act on the promoter region to upregulate gene expression [77,78]. Similarly, the expression of AAT genes is modulated by the interaction between SIRT6 and ATF4 [79]. Under amino acid starvation stress, the protein kinase GCN2/EIF2AK4 phosphorylates eukaryotic translation initiation factor-2α (eIF2α). Phosphorylated eIF2α attenuates global protein synthesis but concurrently enhances the preferential expression of ATF4 and subsequent AAT genes, ensuring that cells can acquire sufficient amino acids to maintain vital functions during nutrient deprivation [80]. By stabilizing TFE3, glucose stress increases the levels of SLC36A1, a lysosomal amino acid transporter that is associated with mTOR activation and the cellular response to glucose starvation [81]. In immune-activated T cells, c-Myc controls the expression of AATs, particularly SLC7A5, inducing a positive feedforward loop to meet the increased amino acid demands for protein synthesis required for optimal T-cell function [82].

Epigenetic modifications are recognized as heritable and reversible alterations in gene activity that do not involve changes in the DNA sequence [83]. DNA methylation is involved in the regulation of several AAT genes, including slc1a2 (EAAT2) [84,85], slc1a3 (GLAST/EAAT1) [85], slc1a4 (ASCT1) [85], slc6a19 (B^0^AT1) [86], slc7a1 (CAT-1) [87], slc7a5 (LAT1) [79,85,88], slc7a8 (LAT2) [89,90], slc7a10 (ASC-1) [85], and slc7a11 (xCT) [85,91]. Histone modifications of AATs include methylation (e.g., H3K4me3 [92] and H3K27me3 [93]) and acetylation [94,95], which occur on N-terminal amino acid residues. As functional RNAs that do not encode proteins, non-coding RNAs (ncRNAs), particularly microRNAs (miRNAs) and long non-coding RNAs (lncRNAs), have been shown to regulate the expression of AAT genes through complementary base pairing with mRNAs. In particular, miRNAs (e.g., miR-23b-3p [96], miR-122 [97,98], miR-194-5p [98], and miR-328-3p [99]) and lncRNAs (e.g., GSTM3TV2) [100] have been shown to be involved in this regulation.

Additionally, 4F2hc/CD98hc is a single-pass transmembrane protein encoded by slc3a2 and plays a critical role in regulating the stability and transport function of AATs such as xCT [101], LAT1 [102], LAT2 [103], and Glut1 [104]. Similar interactions are observed with ACE2, which forms heterodimers with B^0^AT1 (SLC6A19) or SIT (SLC6A20) [105,106], as well as rBAT, which associates with B^0,+^AT (SLC7A9) [107]. Theoretically, any structural or functional changes in these subunits may impact the activity of AATs. For example, the glycosyltransferase B3GNT3 catalyzes the glycosylation of 4F2hc, thereby stabilizing it and enhancing its interaction with xCT. Knockdown of slc3a2 or knockout of B3GNT3 impairs the activity of xCT [108]. Moreover, deficiency of ACE2, a carboxypeptidase that assembles with B^0^AT1 to form a heterodimer, leads to reduced absorption of amino acids, decreased production of intestinal antimicrobial peptides, and increased susceptibility to inflammatory bowel diseases [106].

## 6. Amino Acid Metabolism in Diabetic Cardiomyopathy

In recent years, metabolomics analysis of amino acids has been increasingly utilized in the study of diabetes and diabetic cardiomyopathy (DCM) [109]. Several amino acids and intermediate products have been identified as being correlated with the risk of developing diabetes and diabetic cardiomyopathy (DCM), including BCAAs and AAAs [7,109] Some of these metabolites are considered potential specific biomarkers for DCM [110], although the precise mechanisms underlying their involvement remain to be fully elucidated. Herein, we review the latest research advancements in the metabolism of different amino acids in diabetic cardiomyopathy, including the clinical translation of drug targets. Table 2 summarizes recent preclinical studies related to amino acid metabolism in the context of diabetic cardiomyopathy. These include chemical agents, mimetic compounds, and investigational drugs that have shown potential in attenuating inflammation, oxidative stress, and fibrosis in diabetic cardiac models. Although clinical trials, such as randomized controlled trials (RCTs), are still lacking, these preclinical findings may offer valuable insights for future clinical translation.

### 6.1. Branched-Chain Amino Acids in DCM

BCAAs are the first amino acid group extensively studied in relation to the development of type 2 diabetes and remain the most widely researched to date [7] Branched-chain keto acid dehydrogenase kinase (BCKDK) is a kinase that inhibits BCKDH activity and targets branched-chain keto acid dehydrogenase E1 subunit alpha (BCKDHA), a gene that has been identified as one of the most likely candidate genes for T2DM [149]. PP2Cm, also known as Ppm1k, is the primary activator of BCKDH [150]. Compared with those in control groups, numerous studies have demonstrated significantly elevated plasma levels of BCAAs in both animal models of T1DM and T2DM [151,152] and in patients with these conditions [153,154]. Furthermore, impaired BCAA metabolism results in the accumulation of toxic metabolites [155]. In T1DM, the elevated levels of BCAAs originate from the imbalance between supply (enhanced muscle proteolysis) and consumption (reduced availability of amino group acceptors and diminished BCKDH activity) [151,156,157]. In addition to abnormal metabolic enzymes, genetic variations, overnutrition, and dysbiosis of the gut microbiota are also involved in obesity, insulin resistance and T2DM [156]. Multiple studies have established BCAAs as key markers of diabetes risk [7,8,9] with mechanisms involving interactions with mTOR, as well as increased insulin secretion and pancreatic β-cell depletion caused by BCAA metabolites [7] Recent evidence also suggests a potential link between BCAA metabolites and inflammation, as elevated isoleucine levels were found to correlate with NLRP3 inflammasome activation in patients with nascent metabolic syndrome [158] Such inflammatory activation may represent an underappreciated mechanism contributing to myocardial injury in diabetes-related conditions. Downregulation of BCAA oxidase levels has been shown in experimental models of heart failure and dilated cardiomyopathy [10]. Moreover, increased circulating levels of BCAAs and BCKAs are considered predictive factors for the incidence of coronary heart disease, congestive heart failure, and cardiovascular disease [110]. For example, 3-hydroxyisobutyrate (3-HIB), a valine metabolite, may be highly important for regulating the flexibility of lipid metabolism in the heart [159]. Although alterations in BCAA metabolism have been linked to various heart diseases, with mechanisms including cardiac insulin resistance, myocardial hypertrophy, and impaired myocardial contractility [10,116,156], research on this association in diabetic cardiomyopathy remains insufficient [10]. Given that BCAAs are not major substrates for cardiac fuel supply, it has been proposed that high levels of BCAAs and BCKAs act as signaling molecules to negatively regulate cardiac energy metabolism [10] (Figure 3).

Increased abundance of BCAA-producing bacteria in the gut of diabetic mice [112] and the reduced capacity to degrade BCAAs [160] result in elevated plasma levels of BCAAs. Excess circulating BCAA levels contribute to increased cardiac BCAA levels via the PPARα-FGF21-Zbtb7c-LAT1 axis, leading to cardiac fibrosis and dysfunction [160]. Moreover, dysregulation of the gut microbiota and BCAA metabolism is implicated in diabetes-induced autonomic imbalance, which ultimately leads to cardiac damage [112]. Periostin, an extracellular matrix protein, can be stimulated and upregulated by high-glucose conditions through a TGF-β/Smad-dependent mechanism. In cardiac fibroblasts, impaired BCAA catabolism aggravates cardiac fibrosis via the periostin/NAP1L2/SIRT3 pathway [111]. In rats with T2D, increased levels of AMP deaminase 3 (AMPD3), a negative regulator of BCKDH, impair cardiac energy metabolism. The interaction between AMPD3 and BCKDH offers novel insights into the pathogenesis of DCM [161].

Interventions targeting the metabolic pathway of BCAAs, particularly through the modulation of key enzymes, represent effective strategies for managing diabetes and diabetic cardiomyopathy. The overexpression of PP2Cm reduces BCAA catabolism and oxidative stress, thereby mitigating cardiac ischemia/reperfusion injury [116]. Glucosyringic acid (GA) specifically inhibits periostin expression, suggesting a potential but promising therapeutic approach for DCM [116]. In a mouse model of DCM, pyridostigmine (PYR) enhances vagal nerve activity, restores intestinal microbiota homeostasis, and decreases circulating BCAA levels. Pathologically, PYR alleviates cardiac dysfunction, hypertrophy, and fibrosis [112]. Similarly, the extract from *Portulaca oleracea* L. relieves T2DM by mediating gut microbiota modification and regulating the expression of BCAA-metabolizing enzymes [113]. Moreover, BT2, a selective allosteric inhibitor of BCKDK, has been demonstrated to increase BCAA catabolism, reduce circulating BCAA levels in ob/ob and diet-induced obesity (DIO) mice and significantly potentiate the hypoglycemic effects of metformin [114]. Several additional studies have similarly reported comparable findings, indicating that BT2 improves pathological remodeling and insulin sensitivity in failing hearts [115,116] and mitigates the adverse effects of 3-mercaptopyruvate sulfurtransferase (3-MST) deficiency on heart failure with a reduced ejection fraction (HFrEF) [117]. In addition to its inhibitory effect on BCKDK, the mitochondrial uncoupling property of BT2, which is a lipophilic weak acid, may account for its excellent efficacy in alleviating cardiovascular diseases [118].

The chemical agent BT2, previously mentioned, is not suitable for human application. Sodium phenylbutyrate (NaPB), a drug commonly employed in the management of urea cycle disorder (UCD), binds to the same allosteric pocket as BT2 to inhibit BCKDK [162], thereby reducing BCAA levels in the plasma of both UCD patients and healthy adults [163,164], as well as the BCAA concentration in culture media observed under experimental conditions [119,165]. Chronic exposure to elevated BCAA levels impairs cellular IRS1/AKT signaling pathways, while the administration of NaPB enhances AKT activation, which is posited as one potential mechanism by which NaPB improves insulin sensitivity [165]. In the liver, fibrates can suppress the expression of the BCKDK gene, thereby specifically enhancing the catabolism of BCAAs and reducing the plasma levels of BCAAs in both rodents and humans, as reported by Vanweert et al. [166]. In contrast to allosteric inhibitors, angiotensin II type 1 receptor (AT_1_R) blockers such as valsartan act as ATP-competitive inhibitors of BCKDK, increasing BCKDH activity in the rat liver and reducing plasma BCAA concentrations. Similar inhibitory effects on BCKDK have also been observed with candesartan and irbesartan [167].

Certain antidiabetic medications, including glucagon-like peptide-1 (GLP-1) receptor agonists and sodium-dependent glucose transporters 2 (SGLT-2) inhibitors, exhibit significant cardiovascular protective effects. In diabetic mice, empagliflozin has been shown to improve left ventricular diastolic function, an effect attributed to the downregulation of ryanodine receptor (RyR) phosphorylation and reduced spontaneous calcium release from the sarcoplasmic reticulum during diastole but not related to changes in cardiac BCAA metabolism [168]. Nevertheless, another study investigating the effects of empagliflozin in diabetic mice demonstrated that empagliflozin may mitigate DCM-associated myocardial injury by promoting BCAA catabolism and inhibiting the mTOR/p-ULK1 pathway to increase autophagy [120]. Moreover, research indicates that dapagliflozin exerts anti-inflammatory and fibrosis-lowering effects independent of its hypoglycemic action, suggesting the presence of alternative therapeutic targets [169,170]. Tirzepatide, a dual agonist of gastric inhibitory polypeptide (GIP) and GLP-1 receptors, stimulates the catabolism of BCAAs and BCKAs in brown adipose tissue (BAT), which potentially explains the observed improvements in insulin resistance and reductions in systemic BCAA levels in obese diabetic mice [121]. In contrast to SGLT 2 inhibitors, metformin may increase circulating BCAA and BCKA levels through the AMPK-induced inhibitory phosphorylation of BCKDHA [114,171], which could diminish the cardioprotective benefits of its hypoglycemic effects. In fact, recent studies have highlighted that cardiometabolic comorbidities such as diabetes mellitus not only exacerbate myocardial ischemia-reperfusion injury but also attenuate the efficacy of classical cardioprotective strategies, possibly due to impaired redox signaling and altered cellular stress responses, underscoring the need for tailored interventions in diabetic hearts [172,173].

In addition to pharmacological interventions, endurance training promotes the oxidation of BCAAs and enhances the activity and gene expression of BCKDH; intermittent protein restriction (IPR) exerts positive metabolic effects independent of BCKDH activity and mitigates metformin-induced elevations in plasma BCAA and BCKA levels; cold exposure increases the capacity of mitochondria in BAT to utilize and clear BCAAs [114,166,174,175]; these might be potential alternative strategies for reducing plasma BCAA levels. Furthermore, by targeting the endosomal mTOR-ATPase axis, supplementation with specific amino acids (leucine, lysine, arginine) improves the cardiac insulin resistance and contractile dysfunction induced by lipid overload, with the amino acid transporter SLC38A9 also involved in these beneficial effects [124].

### 6.2. Aromatic Amino Acids in DCM

Similar to BCAAs, the levels of AAAs are regarded as predictors of T2MD [8,9] Interestingly, elevated BCAA levels have been shown to increase AAA levels through competitive inhibition of LAT1 [176].

Phenylalanine levels are regulated by tetrahydrobiopterin-dependent phenylalanine hydroxylase (PAH). Elevated plasma levels of phenylalanine contribute to age-related cardiac aging, characterized by progressive myocardial hypertrophy and interstitial fibrosis and accompanied by diastolic and systolic dysfunction. Administration of tetrahydrobiopterin or dietary restriction of phenylalanine can reverse the age-related elevation of phenylalanine levels and associated cardiac abnormal changes [11]. In diabetic mouse, a significant reduction in tryptophan metabolites and an increase in phenylalanine levels were observed. Meanwhile, rice wine polyphenols (RWPH) and rice wine polypeptides (RWPE) within Chinese rice wine improved these metabolic changes [122]. Administration of the ethanol extract of *S. fusiforme* to a similar mouse model may enhance glucose tolerance by reducing intestinal BCAA and AAA levels, while also alleviating pathological remodeling in cardiac tissue [123].

Several studies have demonstrated a significant increase in the density and number of sympathetic nerve fibers in diabetic hearts, which are corroborated by elevated levels of norepinephrine (NE) and tyrosine hydroxylase (TH) in the myocardium of T2MD animal models [125,126,127]. The activation of the renin-angiotensin-aldosterone system (RAAS) results in increased NADPH oxidase activity, which may directly promote cardiac fibrosis by triggering the TGF-β1/Smad 2/3 signaling pathway [177]. Treatment with SGLT2 inhibitors (e.g., dapagliflozin) and GIP-1 receptor agonists (e.g., liraglutide), exercise, *Stevia rebaudiana* (R) extract, and RAAS blockers (e.g., enalapril and losartan) significantly reduces myocardial NE and TH levels, suggesting that attenuation of sympathetic nerve activity may be a key mechanism underlying the cardioprotective effects observed in DCM [125,126,127]. Correspondingly, bilateral percutaneous renal sympathetic denervation improved heart failure and cardiac remodeling in animal models of DCM, further supporting this hypothesis [177,178].

Abnormal tryptophan metabolism has been implicated in the pathogenesis of several cardiovascular diseases, including heart failure [12,13]. Tryptophan undergoes hydroxylation and decarboxylation to form serotonin (5-HT), which, upon binding to the 5-HT2B receptor, can protect the mitochondria in cardiomyocytes from damage. In diabetes, elevated levels of 5-HT in the gut and serum, coupled with reduced expression of cardiac 5-HT2B receptors, are observed and closely associated with the development of DCM. *Lactobacillus plantarum* and insulin mitigate cardiac apoptosis and fibrosis in diabetic mice by reversing these alterations [128]. Moreover, tryptophan serves as a critical substrate for de novo NAD^+^ synthesis, and cardiac NAD^+^ redox imbalance exacerbates diabetic cardiomyopathy [179]. Increased expression of the key enzyme ACMSD in myocardial endothelial cells impairs NAD^+^ synthesis and increases the risk of cardiac diastolic dysfunction. Inhibition of ACMSD may improve DCM, potentially through activation of the Sirt1/eNOS pathway [13].

### 6.3. Other Amino Acids in DCM

In addition to the extensively studied branched-chain amino acids (BCAAs) and aromatic amino acids (AAAs), the associations between the metabolism of other categories of amino acids and diabetic cardiomyopathy have been investigated to varying degrees in recent years. Nevertheless, existing studies on several amino acids such as threonine and histidine remain limited. Future investigations should focus on exploring the feasibility and potential mechanisms in these uncharted areas to advance our understanding and therapeutic approaches.

#### 6.3.1. Glycine

As precursors to the natural antioxidant glutathione, glycine and glutamic acid levels in the cardiac tissues of rats with DCM are significantly decreased, which may partially explain the link between impaired glutathione synthesis and myocardial injury resulting from oxidative stress [129]. Glycine and serine are interconverted via serine hydroxymethyltransferase (SHMT), whereas disrupted glucose metabolism in diabetic patients leads to serine deficiency and subsequent glycine depletion [176]. Erzhi Pill, a traditional Chinese medicine, effectively improves the metabolism of glycine and glutamate in the cardiac tissue of diabetic patients [129], suggesting a novel therapeutic strategy for DCM. Glycine supplementation ameliorates the integrity of mitochondria-associated endoplasmic reticulum membranes (MAMs) and insulin signaling in liver cells in vitro but does not confer overall metabolic benefits in mice fed high-fat and high-sugar diets [180]. Future studies should investigate the long-term effects of glycine supplementation on T2DM and associated complications.

#### 6.3.2. Serine

In diabetes, as mentioned earlier, depletion of precursors caused by glycolysis increased hepatic and renal consumption due to gluconeogenesis, and decreased expression of enzymes regulating de novo synthesis, coupled with upregulated expression of catabolic enzymes, collectively contribute to serine deficiency [176,181]. Serine deficiency impairs sphingolipid synthesis and leads to the accumulation of neurotoxic deoxysphinganines, which may play a role in the pathogenesis of diabetic neuropathy, including cardiac autonomic neuropathy [176]. In diabetic models, dietary serine restriction in conjunction with a high-fat diet (HFD) accelerates the development of neuropathy in diabetic mice, whereas dietary serine supplementation can mitigate disease progression [181]. Furthermore, serine deficiency is associated with elevated homocysteine (Hcy) levels, likely due to the reduced availability of key methyl donors and impaired activity of cystathionine β-synthase [176]. The implications of elevated Hcy levels will be discussed in subsequent sections.

#### 6.3.3. Methionine

Reduced levels of insulin-like growth factor 1 (IGF-1) and vitamin B12 are observed in patients with T1DM, and a decrease in IGF-1 may contribute to the pathogenesis of DCM. Research indicates that high-dose oral supplementation with vitamin B12 prevents and reverses signs of diabetic cardiomyopathy, likely through direct clearance of reactive oxygen species (ROS) and restoration of SAMe-DNMT-SOCS1/3-IGF-1 signaling [130]. Elevated serum Hcy is considered a risk factor for cardiac fibrosis in diabetes [131], which is partly mediated by differential expression of miRNAs and changes in β-adrenergic signaling transduction [182]. In DCM mice with hyperhomocysteinemia (HHcy), folic acid treatment improved HHcy but did not alleviate the progression of DCM [183]. Moreover, ginger extract reduces cardiac fibrosis and elevates plasma Hcy levels in diabetic rats [131]. Moreover, Hui Tao et al. investigated the mechanism by which DNMT1-mediated methylation of the androgen receptor (AR) triggers homocysteine-induced autophagy in cardiac fibroblasts, suggesting potential roles for AR and DNA methyltransferase 1 (DNMT1) as novel biomarkers of diabetic cardiac fibrosis [184]. S-Adenosylhomocysteine hydrolase (SAHH) catalyzes the hydrolysis of S-adenosylhomocysteine (SAH); elevated plasma levels of SAH caused by the inhibition of SAHH induce endothelial dysfunction. By means of ultrasound-targeted microbubble technology with cationic microbubbles (CMBs) as carriers, delivery of the SAHH gene to cardiomyocytes improved ventricular function in DCM rats, in which the activation of the AMPK/FOXO3/SIRT3 signaling pathway may play a critical role [185].

#### 6.3.4. Cysteine

Hydrogen sulfide (H_2_S), the third gasotransmitter identified to possess signal transduction capabilities, is generated primarily in vivo through the catabolism of cysteine. The physiological roles of endogenous H_2_S have been reviewed [49] and described previously. In recent years, the sources and supplementation of H_2_S have emerged as a research focus for elucidating the relationship between cysteine metabolism and DCM. For example, the novel endogenous H_2_S modulator S-propargyl-cysteine activates insulin receptor signal transduction, thereby exerting beneficial effects on DCM [132]. Exogenous H_2_S supplementation in the form of sodium hydrosulfide (NaHS) prevents lipid deposits in cardiomyocytes by increasing the degradation of sterol regulatory element-binding protein 1 (SREBP1), inhibiting SREBP1 nuclear translocation [133], and regulating Parkin-dependent mitophagy by promoting the S-sulfhydration of ubiquitin-specific protease 8 (USP8) [134], thereby ameliorating DCM. In addition to H_2_S, which is catalyzed by alanine aminotransferase (AST, also known as AAT), cysteine can be converted into another gaseous signaling molecule, endogenous sulfur dioxide (SO_2_), which inhibits the autophagic apoptosis of cardiomyocytes and improves myocardial fibrosis in rats with T2D through the PI3K/AKT pathway [135].

#### 6.3.5. Glutamic Acid and Glutamine

The ratio of glutamic acid to glutamine is considered an indicator of overall energy metabolism, and the imbalance between them may lead to the development of T2DM, with potential mechanisms including oxidative damage, dysfunction, and limited neogenesis of β cells [186]. Previous studies have identified several signaling pathways regulated by glutamine metabolism as key therapeutic targets for DCM. For example, piceatannol [136], empagliflozin [137,138], the angiotensin receptor-neprilysin inhibitor (ARNI) LCZ696 [139] and pioglitazone and curcumin [140] relieve diabetes-induced myocardial oxidative damage and cardiac fibrosis by modulating the Nrf2 and/or NF-κB signaling pathways. Additionally, sulforaphane prevents ferroptosis and associated DCM through activation of the AMPK/Nrf2 pathway [141]. Vitamin D downregulates the expression of the AGE cellular receptor (RAGE) gene and O-glycosylation mediated by the hexosamine pathway and reduces cardiac NF-κB activity, thereby alleviating diabetic cardiomyopathy [142]. These discoveries provide new ideas for the prevention and treatment of DCM in terms of oxidative stress, inflammation, and apoptosis.

#### 6.3.6. Lysine

The currently known degradation pathways of L-lysine include the saccharopine pathway and the pipecolic acid pathway. The former is regarded as the predominant metabolic route, and the detailed mechanisms of the latter remain to be fully elucidated. In the saccharopine pathway, lysine and 2-oxoglutaric acid (OG), derived from the tryptophan metabolic pathway, are enzymatically converted into saccharopine [187]. Saccharopine has been identified as a mitochondrial toxin, and its levels are significantly elevated in patients with DCM compared with healthy controls and those with T2DM without myocardial injury, suggesting a potential link among disruption of the lysine metabolic pathway, mitochondrial dysfunction, and the onset of DCM [110]. The level of 2-aminoadipic acid (2-AAA), a downstream metabolite in the saccharopine pathway, is associated with obesity and metabolic syndrome and can predict the future risk of T2DM. Interestingly, an analysis by Cristina Razquin et al. revealed that higher lysine levels were correlated with an increased future risk of CVD exclusively in diabetic patients [188].

#### 6.3.7. Arginine

Nitric oxide (NO) synthesis is a significant metabolic pathway of arginine. In the presence of adequate tetrahydrobiopterin (BH4) as a cofactor and L-arginine as a substrate, endothelial nitric oxide synthase (eNOS) catalyzes the conversion of L-arginine to NO. In T2DM, increased conversion of BH4 to BH2 leads to eNOS uncoupling, resulting in electron transfer to oxygen and consequently increased production of reactive oxygen species (ROS) and reduced NO generation. Additionally, T2DM enhances arginase activity and increases arginine consumption. The combined supplementation of sepiapterin, the precursor of BH4, and L-citrulline, the precursor of L-arginine, improved the status of diabetic cardiomyopathy in T2DM mice [143]. Moreover, previous studies have demonstrated that arginase inhibitors can reverse the exacerbation of cellular oxidative stress induced by high-glucose stimulation [189]. NG-dimethyl-L-arginine (ADMA), a natural L-arginine analog, can be metabolized by dimethylarginine dimethylaminohydrolase 2 (DDAH2), competitively inhibiting eNOS and leading to its uncoupling. The overexpression of DDAH2 improves myocardial fibrosis and cardiac function by activating the DDAH/ADMA/eNOS/NO pathway, thereby delaying the progression of DCM [190].

Polyamines (PAs) represent another critical group of metabolic products of arginine, including putrescine, spermidine, and spermine. Metabolomic analysis has confirmed that spermine mitigates myocardial injury in DCM by modulating the metabolic pathways of lipids and amino acids, with acyl-CoA thioesterase 1 (Acot1) potentially serving as one of the key targets [191]. In DCM, the expression levels of various polyamines and metabolic enzymes, such as ornithine decarboxylase (ODC), are altered. Therapy with polyamines attenuates diabetic cardiomyopathy by inhibiting the downregulation of calcium-sensing receptors [144] alleviating endoplasmic reticulum stress, and modulating Wnt signaling pathways [145]. Tyler N Kambis et al. reviewed the changes in miRNAs in hearts with DCM, which regulate cell death, oxidative stress, myocardial hypertrophy and fibrosis, thereby influencing cardiac remodeling [192]. Notably, miRNAs also play crucial regulatory roles in polyamine synthesis, suggesting a potential new approach for targeting polyamine metabolism to improve DCM [192].

The application of L-arginine to the neonatal rat cardiomyocyte cell line H9c2 protects human serum albumin (HSA) from glycosylation, ensuring the proper function of nuclear factor erythroid 2-related factor 2 (Nrf-2), which suggests the potential of L-arginine in mitigating accelerated glycosylation and oxidative stress-associated DCM [146]. Moreover, L-arginine supplementation prevents the development of DCM in mice with T2DM by enhancing mitochondrial function [193]. The combined administration of L-arginine and the cannabinoid 2 (CB2) receptor agonist β-caryophyllene has also been revealed to alleviate inflammation-induced cardiac dysfunction by downregulating NF-κB expression in diabetic hearts [147].

#### 6.3.8. Alanine

Elevated alanine aminotransferase (ALT) levels are observed in high-fat diet (HFD)-induced T2DM mice, suggesting progressive hepatic injury. Concurrently, echocardiographic assessments reveal manifestations of cardiomyopathy characterized by a reduced left ventricular ejection fraction (EF%), fractional shortening (FS%), and fractional area change (FAC%) [194]. In diabetes, the relationship between elevated ALT levels and the development of mild cardiomyopathy, in which metabolic disturbances secondary to liver damage may play a contributory role, remains to be elucidated.

#### 6.3.9. Aspartic Acid and Asparagine

As a serum biomarker indicative of cardiac injury, the level of aspartate aminotransferase (AST) relatively increases when myocardial injury occurs. Several plant-derived compounds that act on DCM animal models have been shown to reduce elevated serum AST levels and exhibit cardioprotective effects against hyperglycemia-induced damage, including *Lycium chinense* leaf extract [195], *Artemisia vulgaris* extract [196], olive leaf extract [197], ginger extract [198], sodium houttuyfonate [199], cinnamon [200], quercetin 4′-O-glucoside [148], and thymoquinone [201]. Similar benefits were observed in moderate-to-high intensity endurance exercise [202] and β-aminoisobutyric acid treatment [201]. A study based on the China Cardiometabolic Disease and Cancer Cohort (4C) revealed that during the stage of normal glucose tolerance, abnormal asparagine metabolism may also be indicative of an increased risk of future T2DM development [203]. Moreover, in T1DM, reduced levels of asparagine and glutamine are associated with the progression of cardiovascular autonomic neuropathy induced by diabetes [204]. We hypothesize that, given the close relationship between asparagine and glutamine and the tricarboxylic acid cycle, metabolic disturbances may impact the glycolytic pathway, which is essential for peripheral nerve function; however, more research is needed to elucidate further the mechanisms involved.

#### 6.3.10. Proline

Metabolic syndrome (MetS) is a preclinical high-risk state that is associated with the development of T2DM and cardiovascular diseases. According to research from the China Suboptimal Health Cohort, the metabolism of proline, arginine, and glutathione is influenced in individuals with metabolic syndrome [205]. Fibrosis is a key pathological feature of DCM. Elevated expression of glutamyl-prolyl-tRNA synthetase (EPRS), the sole enzyme in the mammalian cytoplasm responsible for catalyzing the synthesis of prolyl-tRNA, leads to increased formation of prolyl-tRNA, which directly enhances the translation of proline-rich extracellular matrix proteins, thereby promoting myocardial fibrosis under pathological conditions. The inhibition of EPRS can alleviate the profibrotic effects of proline-rich collagens and TGF-β, potentially ameliorating cardiac fibrosis in diabetic cardiomyopathy [206].

## 7. Conclusions and Perspectives

Although the dysregulation of glucose and lipid metabolism remains the primary initiating factor of DCM, the crosstalk among metabolic pathways and signaling networks underscores the indispensable role of amino acid metabolism in maintaining homeostasis and supporting growth and development. In fact, amino acid metabolism critically drives DCM progression through multifaceted mechanisms, such as (i) the vicious cycle between BCAA metabolism disorders and insulin resistance; (ii) arginine metabolism imbalances and microcirculation disorders; (iii) disturbances in sulfur-containing amino acid metabolism and oxidative stress damage; and (iv) glutamine metabolism reprogramming and the energy crisis. This review systematically summarizes the fundamental pathways of amino acid metabolism, highlights several critical regulatory targets, and aims to provide deeper insights for the understanding and management of diabetic cardiomyopathy.

Here, we discuss various pharmacological agents that have demonstrated efficacy in animal models and in vitro cells by reducing cardiac fibrosis and ventricular pathological remodeling and alleviating diastolic and systolic dysfunction to ameliorate DCM. More significantly, certain drugs and amino acid supplements may offer clinical benefits for specific patient groups. In a randomized controlled trial, NaPB increased peripheral insulin sensitivity in T2DM patients and simultaneously reduced the levels of BCAAs and glucose [207]. Fibrates, as classic lipid-lowering drugs used to reduce cardiovascular risk, have also been shown to improve insulin sensitivity and BCAA metabolism [166]. Although the specific mechanism remains unclear, these findings imply that fibrates may protect diabetic hearts through the glucose, lipid, and amino acid metabolic pathways. Similarly, in addition to blocking AT1R, the inhibitory effect of valsartan on BDKDK may represent another pharmacological mechanism for sartans in the treatment of diabetic heart diseases. In elderly populations, GlyNAC (a combination of glycine and N-acetylcysteine) supplementation improves oxidative stress, endothelial dysfunction, and insulin resistance [208]. Additionally, glutamine supplementation stimulates insulin secretion via increased GLP-1 production, thereby lowering blood glucose levels and ameliorating cardiac risk factors in T2DM patients [186,209]. However, excessive glutamine may exacerbate myocardial injury through increased glutamate production or O-GlcNAcylation [210], which reflects the complexity of amino acid metabolic regulation. Future research should build upon preclinical findings to validate the efficacy and safety of pharmacological agents across both clinical trial settings and diverse patient populations.

Clearly, the management of diabetes complications, particularly diabetic cardiomyopathy, represents a complex and long-term systemic endeavor that requires multidisciplinary collaboration, sustained government support, and the integration of societal resources. Despite significant advancements in understanding the pathophysiology of diabetes and its cardiovascular sequelae, numerous challenges remain unresolved. In this context, it is anticipated that emerging research on amino acid metabolism—including the integration of artificial intelligence-empowered metabolomic profiling and advanced multimodal cardiac imaging techniques—will contribute significantly to improving the prevention, diagnosis, and treatment of diabetes and its cardiac complications [205,211,212,213].

Amino acid metabolism plays a pivotal and multifaceted role in the pathogenesis of diabetic cardiomyopathy. Beyond the classical view of glucose and lipid dysregulation, targeting specific amino acid pathways offers novel opportunities for diagnosis, risk stratification, and therapeutic intervention. Advancing our understanding of these mechanisms may pave the way for more precise and individualized management strategies for patients with diabetes and cardiovascular complications.

## Figures and Tables

**Figure 1 biomolecules-15-00916-f001:**
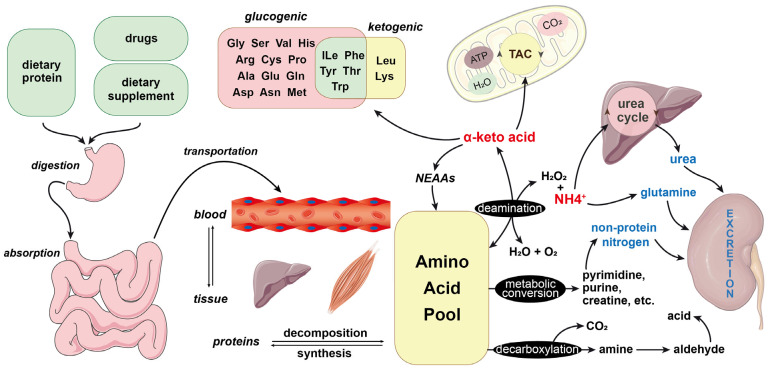
Pathways of amino acid metabolism. Most amino acids undergo catabolic processes in the liver, primarily through deamination to form α-keto acids and ammonia (NH3/NH4^+^). The resulting α-keto acids are oxidized in the mitochondria for energy production, converted into glucose and fatty acids, or resynthesized into NEAAs. Ammonia is predominantly detoxified via the ornithine cycle to form urea and ultimately excreted by the kidneys. Moreover, certain amino acids can be decarboxylated to yield carbon dioxide and amines. The arrows in the figure represent either the transport of substances or the course of biochemical processes.

**Figure 2 biomolecules-15-00916-f002:**
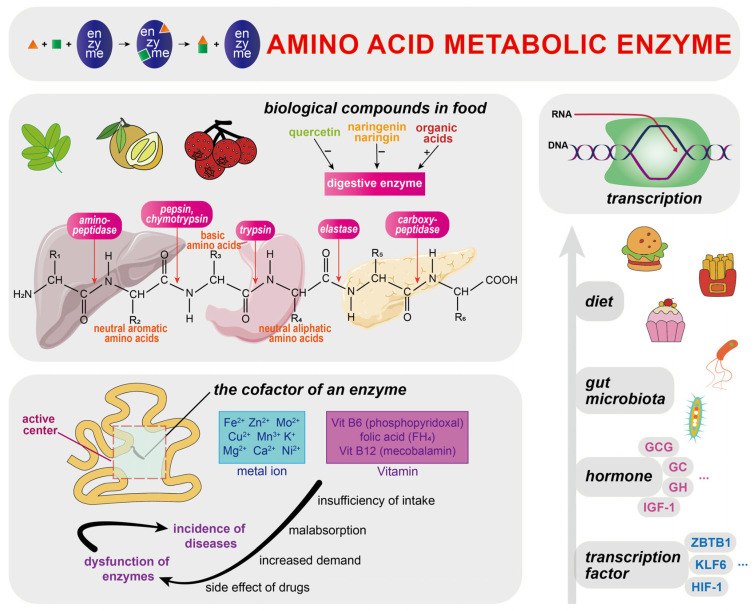
Regulation of amino acid metabolic enzymes. Enzymes involved in amino acid metabolism are regulated by a variety of factors. Bioactive compounds in food, such as quercetin and organic acids, can inhibit or activate the activity of different digestive enzymes. Cofactor dysfunction may lead to impaired enzyme activity and disease pathogenesis. In addition, the transcription of enzymes is regulated by factors such as diet, the gut microbiota, hormone levels, and transcription factors. + and − represent the activation and inhibition of digestive enzyme activity, respectively.

**Figure 3 biomolecules-15-00916-f003:**
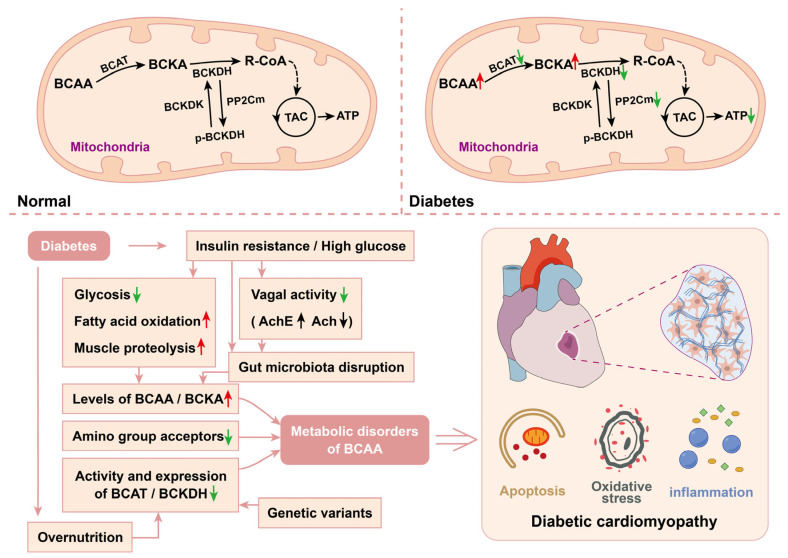
Branched-Chain Amino Acids Metabolism in DCM. Under physiological conditions, BCAAs undergo catabolism via a series of enzymatic reactions in the mitochondria of skeletal muscle, myocardium, and hepatic tissues. In diabetes, the interplay of impaired glucose/lipid metabolism, vagal dysfunction, and gut microbiota disruption alters the status of substrates and enzymes within the myocardial BCAA metabolic pathway, which contributes to the pathogenesis of DCM. Mechanistically, autophagy dysregulation, oxidative stress, and inflammatory responses collectively drive myocardial hypertrophy and fibrosis, ultimately manifesting as impaired cardiac function. An upward arrow indicates an increase in a substance or the promotion of a biological process, while a downward arrow indicates a decrease in a substance or the inhibition of a biological process.

**Table 1 biomolecules-15-00916-t001:** Classification, derivatives, and main degrading enzymes of amino acids.

Name (Abbreviation, Symbol)	Nutritional Classification	Metabolic Transformation	Important Nitrogenous Derivative	Main AADEs (Abbreviation)
Alanine (Ala, A)	NEAA	Glucogenic	-	Glutamic pyruvate transaminase (GPT)
Cysteine (Cys, C)	NEAA	Glucogenic	Taurine	Cysteine dioxygenase 1 (CDO1)
Aspartic acid (Asp, D)	NEAA	Glucogenic	Purine base, pyrimidine base	Glutamic-oxaloacetic transaminase 1 (GOT1)
Glutamic acid (Glu, E)	NEAA	Glucogenic	GABA	Glutamate dehydrogenase 1 (GLUD1)
Phenylalanine (Phe, F)	EAA	Glucogenic and ketogenic	CA, thyroxine, melanin	Phenylalanine hydroxylase (PAH)
Glycine (Gly, G)	NEAA	Glucogenic	Purine base, porphyrin, creatine, creatine phosphate	Serine hydroxymethyltransferase 1 (SHMT1)
Histidine (His, H)	EAA	Glucogenic	Histamine	Histidine ammonia-lyase (HAL)
Isoleucine (Ile, I)	EAA	Glucogenic and ketogenic	-	Branched-chain amino acid transaminase (BCATc, BCATm)
				Branched-chain keto acid dehydrogenase (BCKDH)
Lysine (Lys, K)	EAA	Ketogenic	Crotonyl-CoA	Glutaryl-CoA dehydrogenase (GCDH)
Leucine (Leu, L)	EAA	Ketogenic	-	see Isoleucine
Methionine (Met, M)	EAA	Glucogenic	Spermidine, spermine, creatine, creatine phosphate	Methionine adenosyl transferase 1A (MAT1A)
Asparagine (Asn, N)	NEAA	Glucogenic	-	Asparaginase (ASPG)
Proline (Pro, P)	NEAA	Glucogenic	-	Proline dehydrogenase 1 (PRODH1)
Glutamine (Gln, Q)	NEAA	Glucogenic	Purine base	Glutaminase 1 (GLS1)
				Glutaminase 2 (GLS2)
Arginine (Arg, R)	NEAA	Glucogenic	NO, Creatine, creatine phosphate	Arginase 1 (ARG1)
Serine (Ser, S)	NEAA	Glucogenic	-	Serine dehydratase (SDS)
Threonine (Thr, T)	EAA	Glucogenic and ketogenic	-	Serine dehydratase like (SDSL)
Valine (Val, V)	EAA	Glucogenic	-	see Isoleucine
Tryptophan (Trp, W)	EAA	Glucogenic and ketogenic	5-HT, nicotinic acid	Tryptophan 2,3-dioxygenase (TDO2)
				Aminocarboxymuconate semialdehyde decarboxylase (ACMSD)
				Tryptophan hydroxylase 1 (TPH1)
Tyrosine (Tyr, Y)	NEAA	Glucogenic and ketogenic	CA, thyroxine, melanin	Tyrosinase (TYR)
				Tyrosine hydroxylase (TH)
				Tyrosine aminotransferase (TAT)

**Table 2 biomolecules-15-00916-t002:** Preclinical research on amino acid metabolism in diabetic cardiomyopathy.

Target	Drug	Model	Main Findings	Refs
BCAT2, PP2Cm	Glucosyringic acid (GA)	Male C57BL/6 J mice induced by HFD and SFZ	GA restored normal BCAA metabolism in diabetic mouse heart via targeting and inhibiting the periostin/NAP1L2/SIRT3 axis	[111]
BCAT2, PP2Cm, BCKDK, BCKDH	Pyridostigmine	Male C57BL/6 J mice induced by HFD and SFZ	Pyridostigmine improved disrupted BCAA metabolic enzymes and intestinal microbiota homeostasis, and enhanced vagal nerve activity	[112]
PP2Cm, BCKDK, BCKDH	*Portulaca oleracea* L. extracts (PE)	Mice induced by HFD and STZ	PE improved disrupted BCAA metabolic enzymes and intestinal microbiota homeostasis	[113]
BCKDK	3,6-dichlorobenzo[b] thiophene-2-carboxylic acid (BT2)	Male ob/ob mice, wild-type C57BL/6 J mice, leptin gene mutant ob/ob mice and BCKDK-Alb cre+ mice, induced by HFD	BT2 inhibited BCKDK, reduced the level of p-BCKDH and thus enhanced BCAA catabolism, and potentiated the hypoglycemic effect of metformin	[114]
BCKDK	BT2	Male C57BL/6 N mice performed with TAC	BT2 enhanced BCAA catabolism, and improved TAC-induced cardiac contractile dysfunction and pathological state	[115]
BCKDK	BT2	C57BL/6 mice induced by HFD and SFZ	BT2 enhanced BCAA catabolism, ameliorated the impaired heart function and reduced the infarction area in diabetic mice with myocardial infarction	[116]
BCKDK	BT2	3-MST KO mice and C57BL/6 J mice performed with TAC	BT2 enhanced BCAA catabolism and ameliorated the severity of TAC-induced heart failure in 3-MST KO mice	[117]
The oxidative respiration process within the mitochondria	BT2	NRVMs and iPSC-derived cardiomyocytes	Independent of the inhibitory effect on BCKDK, BT2 decreased mitochondrial membrane potential, increased proton conductance across the mitochondrial inner membrane, and reduced the production of ROS	[118]
IRS1/Akt signaling pathway	Sodium Phenylbutyrate (NaPB)	Murine C2C12 myoblasts, passage 6–8, treated with elevated (4×) media BCAA concentrations	Under high BCAA conditions, NaPB treatment elevated Akt and AS160 phosphorylation, while decreased glycogen synthesis and BCAA concentrations	[119]
PP2Cm and mTOR/p-ULK1 signaling pathway	Empagliflozin	Male KK-Ay mice aged 8 weeks induced by HFD	Empagliflozin promoted BCAA degradation through the upregulation of PP2Cm and inhibited mTOR/p-ULK1 to enhance autophagy	[120]
BCAA/BCKA metabolism in BAT	Tirzepatide	Male C57BL/6 J mice induced by HFD	Tirzepatide stimulated catabolism of BCAAs/BCKAs in BAT, as demonstrated by increased BCAA/BCKA-derived metabolites	[121]
PAH in livers	Tetrahydrobiopterin	Global p21^−/−^ mice backcrossed to C57BL/6 J background for at least 10 generations	In naturally aged mice, consistent with siRNA-mediated p21 knockdown, tetrahydrobiopterin treatment restored healthy cardiac structure and function through reviving hepatic PAH activity and normalizing plasma phenylalanine levels	[11]
Gut microbiota	Rice wine polyphenols and polypeptides within Chinese rice wine	Male db/db and db/m mice	Functional components of Chinese rice wine provided a cardioprotective effect against DCM via increasing tryptophan metabolism-associated metabolites and reducing serum phenylalanine by modulating the composition and metabolic function of the gut microbiota	[122]
Gut microbiota	Ethanol extract of *S. fusiforme* (EE)	Male ICR mice aged 8 weeks induced by HFD and SFZ	EE altered the composition of gut microbiota, reduced the levels of BCAAs and AAAs, and improved glucose tolerance as well as pathological changes in the heart	[123]
mTORC1-v-ATPase axis, adaptor protein Ragulator, and SLC38A9	Specific cocktail of amino acids (lysine/leucine/arginine)	Male Lewis rats induced by HFD, and cardiomyocyte models (aRCMs, HL-1 and hiPSC-CM)	Lysine/leucine/arginine stimulated mTORC1-v-ATPase axis, reinternalized CD36, and reduced cardiac lipid uptake	[124]
Myocardial NE and TH	Liraglutide and dapagliflozin	Male SD rats induced by HFD and SFZ	Both liraglutide and dapagliflozin significantly reduced TH density and myocardial NE contents, and dapagliflozin exhibited more reduction than liraglutide	[125]
Myocardial TH	*Stevia Rebaudiana* (R) extracts	Male SD rats induced by HFD and SFZ	*Stevia* R extracts significantly attenuated myocardial TH density	[126]
Myocardial NE and TH	RAAS blockers (enalapril and losartan)	Male SD rats induced by HFD and SFZ	Blockade of RAAS attenuated myocardial TH density and NE contents	[127]
5-HT and its cardiac receptor (5-HT2B receptor)	*L. plantarum* and inulin	Male Wistar rats induced by HFD and SFZ	Increase in intestinal and serum 5-HT as well as decrease in cardiac 5-HT and 5-HT2B receptor were observed in diabetic rats, which were reversed by *L. plantarum* and insulin administration	[128]
Amino acid metabolism and AMPK and PI3K/Akt/FoxO3a signaling pathways in the heart tissue	Erzhi Pill	Male SD rats induced by HFD and SFZ	Erzhi Pill balanced amino acid metabolism similar to glutamic acid and glycine, and regulated the AMPK and PI3K/Akt/FoxO3a signaling pathways	[129]
SAMe and DNMT-SOCS1/3-IGF-1 signaling	Vitamin B12	Male C57BL/6 J mice carrying *Elmo1H/H* and *Ins2Akita/+* genes	High oral dose of vitamin B12 normalized the decreased levels of SAMe and DNMTs, modulated oxidative stress, and improved the echocardiographic indices	[130]
Hcy, etc.	Ginger extract	Male Wistar rats induced by SFZ	Ginger extract restored the increased levels of Hcy and alleviated heart structural abnormalities	[131]
Cardiac CSE and H_2_S as well as insulin receptor and Akt/GSK-3β signaling	S-Propargyl-Cysteine (SPRC)	Male C57BLKS/J db/db mice	SPRC increased CSE expression and H_2_S content, activated cardiac insulin receptor and Akt/GSK-3β signaling	[132]
Synoviolin-1 (SYVN1/Hrd1)	NaHS and the novel hydrogen sulfide-releasing molecule GYY4137	Female db/db mice and HL-1 cells treated with palmitate and oleate	Exogenous H_2_S improved H_2_S levels in cardiomyocytes, prevented LDs formation by restoring SYVN1 S sulfhydration and promoting SREBP1 ubiquitination	[133]
USP8/parkin signaling pathway	NaHS	Male and female db/db mice	Exogenous H_2_S activated USP8 S sulfhydration, promoted parkin-dependent mitophagy and ameliorated cardiac impairment	[134]
PI3K/Akt pathway	Exogenous SO_2_ donor (Na_2_SO_3_/NaHSO_3_)	Male SD rats induced by high-fat high-sucrose diet (HFHSD) and SFZ	SO_2_ activated autophagy to antagonize cardiomyocyte apoptosis and fibrosis by downregulating the excessive activation of PI3K/Akt pathway	[135]
Nrf2/HO-1 and NF-κB pathways	Piceatannol (PIC)	Male SD rats induced by SFZ, and HG-induced H9C2 cardiac myoblasts	PIC suppressed HG-induced NF-κB activation by upregulating Nrf2 and HO-1 expression, and alleviated inflammation and oxidative stress in DCM rats	[136]
Nrf2/ARE signaling pathway	Empagliflozin	Male db/db mice	Empagliflozin inhibited oxidative stress via activating Nrf2/ARE signaling, modulated ketone body metabolism, and improved mitochondrial dysfunction in DCM	[137]
Nrf2/ARE signaling and TGF-β/Smad pathway	Empagliflozin	KK-Ay mice induced by HFD	Empagliflozin attenuated oxidative stress and fibrosis in diabetic heart by activating Nrf2/ARE and suppressing TGF-β/Smad signaling	[138]
JNK/p38 MAPK and NF-κB pathways	LCZ696 (an ARNI)	Male C57BL/6 mice induced by SFZ, and HG-induced H9C2 cardiomyocytes	LCZ696 inhibited inflammation and oxidative stress by suppressing JNK/p38 MAPK phosphorylation and NF-κB nuclear translocation	[139]
CaMKII/NF-κB/TGF-β1 and PPAR-γ signaling pathways	Pioglitazone and curcumin (Pio/Cur)	Male adult SD rats induced by SFZ	Pio/Cur treatment ameliorated DCM in T1DM via inhibition of CaMKII/NF-κB/TGF-β1 and activation of PPAR-γ pathways	[140]
AMPK/Nrf2 pathways	Sulforaphane (SFN)	Engineered cardiac tissue and AMPKα2-KO mice induced by HFD and SFZ	SFN prevented ferroptosis and associated cardiac pathogenesis via AMPK-mediated Nrf2 activation	[141]
RAGE, OGT, and GFAT and NF-κB in heart tissue	Vitamin D	Male SD rats induced by SFZ	Vitamin D alleviated DCM by down-regulating the RAGE expression and HBP-mediated *O*-glycosylation, while reducing NF-κB activity	[142]
BH4/eNOS/NO pathway	Sepiapterin (SEP) and L-citrulline (L-Cit)	db/db mice and HG-induced ECs stimulating I/R or H/R conditions	Coadministration of SEP and L-Cit protected diabetic heart, via improvements in coronary arterial endothelial function, cardiac BH4 concentrations, and eNOS function	[143]
Nrf2-ROS-p53-MuRF1 axis	Spermine	Male Wistar rats induced by SFZ	Exogenous spermine attenuated DCM by suppressing ROS-p53 mediated downregulation of cell membrane calcium-sensitive receptor	[144]
Wnt/β-catenin signaling	Spermine	Male Wistar rats induced by SFZ, and HG-induced CFs from neonatal Wistar rats	Exogenous spermine attenuated myocardial fibrosis by inhibiting ERS and the canonical Wnt/β-catenin signaling pathway	[145]
Nrf2 signaling	L-Arginine	Neonatal rat cardiomyocytes H9c2 (2-1) cell line, incubated with MGO to stimulate glycation	L-Arginine exerted protective effects in DCM due to the inhibition of HSA glycation as well as the activation and nuclear translocation of Nrf2	[146]
NF-κβ pathway	β-caryophyllene (BCP) and L-Arginine (LA)	Male SD rats induced by SFZ	Coadministration of BCP and LA led to a reduction in collagen deposition and cardiac fibrosis via NF-ĸβ inhibition	[147]
PI3K/Akt/Nrf2 pathway	Spiraeoside	HG-induced AC16 cells	Spiraeoside protected HG-stimulated cardiomyocytes through its antioxidant and antiapoptotic activities via the activation of PI3K/Akt/Nrf2 pathway	[148]

## Data Availability

Not applicable.

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
