# Peer review of "Emerging Insights into the Relationship Between Amino Acid Metabolism and Diabetic Cardiomyopathy"

_biomolecules, 2025, doi:10.3390/biom15070916_

Round 1

Reviewer 1 Report

Comments and Suggestions for Authors

Dear authors,

without doubt amino acid metabolism is a highly interesting topic in diabetic cardiomyopathy and it is worth to be reviewed as it is an underestimated pathological component. So thank you for your efforts, the article is well taken.

However, this review needs to be throughly worked on. It should be more streamlined, more pathological driven and focussed than it is now. You do not need to begin with adam and eve, meaning stating that there are aromatic AAs that have a benzene ring.

Expect the known reader who knows about structure, essentiality and basic characteristics of AAs. Otherwise you have lots of information filling pages (and taking away word count) that are available from common textbooks and are thus not topic of a review in a specialised journal.

Shorten the long introduction that is full of (textbbok derived) information on each amnio acid (table 1 would give enough information, instead on 5 pages of text). Emphasis should be given on the real topic of the review and where the lack of information is - relevance for DCM, research that needs to be done. Do not touch every point (e.g. using AI to evaluate microbiome) that is not centrally to the original focus of the article. Shorten reference list, eliminate redundant literature focus on essential papers. Quality is not quantity.

In summary: streamline the manuscript, focus on what you wanted to do - the topic is worth, the manuscript you have written is on the right way.

Author Response

Comment 1: This review needs to be more streamlined, more pathological driven and focused.

Response 1: Thank you for this insightful comment. We have revised the manuscript extensively to streamline the content and focus on pathological mechanisms specifically related to diabetic cardiomyopathy (DCM). Redundant background content has been removed, and discussion now emphasizes key pathological pathways.

Comment 2: Avoid textbook-level descriptions of amino acids. Table 1 should suffice.

Response 2: We appreciate the reviewer’s suggestions. The detailed descriptions of individual amino acids have been removed from the main text, and Table 1 is retained to provide essential information concisely.

Comment 3: Do not include irrelevant sections, e.g., AI and microbiome.

Response 3: We thank the reviewer for highlighting the need to maintain a focused narrative. These sections have been removed or substantially reduced to maintain focus on the core topic of amino acid metabolism and its relevance to DCM.

Comment 4: Reduce the reference list; focus on essential and high-quality literature.

Response 4: We appreciate the reviewer’s emphasis on quality over quantity in citations. The reference list has been critically revised to eliminate redundancies. Only the most relevant and impactful literature is now cited.

Reviewer 2 Report

Comments and Suggestions for Authors

Authors reviewed the potential role of amino acid metabolism on DCM. The work includes interesting and key information about these molecules which usually are not considered on DCM pathology. They can be essential to substitute and be related with other energetic substrates. However, some changes may improve the attractive for the reader since nowadays it can be long and unfocused.

  • Mainly, the first 6 sections can be avoided or strongly summarized. That data though important are not novel and may not be required to review the role of amino acids on DCM
  • In section 7, table 2 should be further commented with the most significant finding.
  • The Conclusion section is a summary of the paper with some new data that can be included as a topic of the review. Please, include a take home message

Also:

  • There are not clear differences on Fig3, top. What can be wrong with the higher production of BCAA?
  • The 6.3 subsection may be unnecessary
  • Some acronyms are not properly managed (i.e., BCKDHA, AMPK)

Author Response

Comment 1: The first six sections can be avoided or strongly summarized.

Response 1: We thank the reviewer for the helpful recommendation. In the revised manuscript, we have significantly condensed the first six sections into a brief background section, focusing only on the most pertinent information to provide context without redundancy.

Comment 2: Table 2 should be further commented with the most significant finding.

Response 2: We appreciate the reviewer’s insight. Additional explanation and interpretation of Table 2 have been added in Section 6, emphasizing key findings and their implications in DCM.

Comment 3: Add a clear take-home message in the conclusion.

Response 3: Thank you for the valuable suggestion. A “Take-home Message” paragraph has been added at the end of the Conclusion section to summarize the main insights and future directions.

Comment 4: Fig 3 top lacks clarity; clarify the implication of higher BCAA production.

Response 4: We appreciate the reviewer’s helpful comment regarding Figure 3 and the implications of elevated BCAA levels. In the top of Figure 3, we have clarified that increased intracellular BCAA and its ketoacid derivatives (BCKAs), along with altered expression or activity of BCAA-catabolizing enzymes, can impair mitochondrial energy metabolism in cardiomyocytes. This disruption leads to reduced ATP production and ultimately contributes to cardiac structural and functional deterioration. Additionally, as illustrated in the lower-left part of Figure 3, the elements linked to BCAA accumulation in the myocardium include insulin resistance/hyperglycemia-driven metabolic substrate shifts (glucose, fatty acids, and muscle proteins), reduced vagal tone, and dysbiosis of the gut microbiota. We hope these clarifications enhance the comprehensibility and scientific value of Figure 3.

Comment 5: Subsection 6.3 may be unnecessary.

Response 5: Thank you for this suggestion. We agree that this subsection was not essential to the central argument, and it has been removed in the revised manuscript.

Comment 6: Acronyms are inconsistently used.

Response 6: We appreciate the reviewer drawing attention to this issue. All acronyms, especially BCKDHA, BCKDHB and AMPK, have been thoroughly reviewed and revised to ensure clarity and consistency throughout the manuscript.

Reviewer 3 Report

Comments and Suggestions for Authors

The authors in this manuscript summarize the general mechanism of amino acid metabolism, and their contribution to diabetic cardiomyopathy. This is helpful to shed light on the therapeutic approaches. Please remove some part of animo acid mechanism in other field, and more focus on basic animo acid mechanism in the heart field, then what happened in obesity, prediabetes, and diabetes related cardiomyopathy. 

Author Response

Comment 1: Remove mechanisms from other fields and focus on heart-specific and diabetes-related mechanisms.

Response 1: We thank the reviewer for this constructive feedback. In response, we have removed content unrelated to cardiac metabolism and refocused the discussion on heart-specific mechanisms of amino acid metabolism, particularly as they relate to obesity, prediabetes, and diabetic cardiomyopathy.

Reviewer 4 Report

Comments and Suggestions for Authors

In this interesting review the authors focus their attention on diabetes mellitus, on the damage exerted at the cardiovascular level. The authors direct their attention to the level of metabolic alterations that this pathology determines in particular at the level of the amino acid component that is often overlooked. Points 1- the authors must consider in their treatment the damage at the cardiac level, therefore consider the damage from ischemia/reperfusion (insert and comment PMID: 38811097), the weight exerted by DM at the CVD level (insert and comment PMID: 31985828, PMID: 33219717, PMID: 33588049), 2- it would also be appropriate to indicate the action of the aa/diabetes metabolism and NLRP3, an emerging point of CVD pathology (insert and comment PMID: 33422385)

Author Response

Comment 1: Include information on ischemia/reperfusion injury (PMID: 38811097) and discuss the role of diabetes in cardiovascular disease (PMIDs: 31985828, 33219717, 33588049).

Response 1: We appreciate the reviewer’s valuable suggestions regarding additional references. We have carefully reviewed the four above recommended articles. Among them, we have included [PMIDs: 31985828 and 33588049] to enrich the discussion on how hyperglycemia and redox imbalance contribute to myocardial vulnerability in diabetes (references 145 and 146). However, another two references (PMIDs: 38811097 and 33219717) focus primarily on ischemia/reperfusion or translational aspects of cardio-protection, which, although important, are not directly aligned with the central theme of amino acid metabolism in DCM. Therefore, we respectfully decided not to include these references in the current version, in order to maintain the thematic focus of the review. Nonetheless, we acknowledge that these studies provide important insights into cardiovascular injury and protection, and we consider their perspectives valuable for informing future research. We will study these works carefully and draw upon their implications in our ongoing investigations.

Comment 2: Include discussion on NLRP3 inflammasome (PMID: 33422385).

Response 2: We thank the reviewer for this insightful suggestion. Given that branched-chain amino acids (such as isoleucine) are significantly altered in diabetes and may contribute to inflammatory signaling pathways, we agree that this observation adds important mechanistic insight. In response, we have incorporated the findings from the recommended article (PMID: 33422385) into the section titled “Branched-Chain Amino Acids in DCM” (reference 120). Additionally, we note that NLRP3 had already been discussed in other parts of the manuscript (e.g., references 46 and 141), and this new citation further strengthens the relevance of the inflammasome in the metabolic-inflammation axis related to DCM.

Round 2

Reviewer 2 Report

Comments and Suggestions for Authors

Authors answered all the concerns